# An improved reduced-order model for pressure drop across arterial stenoses

**Konstantinos G. Lyras**⊕◎\*, **Jack Lee**◎\*

School of Biomedical Engineering & Imaging Sciences, King's College London, London, United Kingdom

◎ These authors contributed equally to this work.
\* konstantinos.lyras@kcl.ac.uk (KGL); jack.lee@kcl.ac.uk (JL)

## Abstract

Quantification of pressure drop across stenotic arteries is a major element in the functional assessment of occlusive arterial disease. Accurate estimation of the pressure drop with a numerical model allows the calculation of Fractional Flow Reserve (FFR), which is a haemodynamic index employed for guiding coronary revascularisation. Its non-invasive evaluation would contribute to safer and cost-effective diseases management. In this work, we propose a new formulation of a reduced-order model of trans-stenotic pressure drop, based on a consistent theoretical analysis of the Navier-Stokes equation. The new formulation features a novel term that characterises the contribution of turbulence effect to pressure loss. Results from three-dimensional computational fluid dynamics (CFD) showed that the proposed model produces predictions that are significantly more accurate than the existing reduced-order models, for large and small symmetric and eccentric stenoses, covering mild to severe area reductions. FFR calculations based on the proposed model produced zero classification error for three classes comprising positive ($\leq 0.75$), negative ($\geq 0.8$) and intermediate ($0.75 - 0.8$) classes.

## Introduction

The relationship between the flow and pressure of blood across a stenosed artery is central to assessing the functional severity of occlusive arterial disease. Since the 1990s when the interventional cardiology community turned their attention to functional assessments, influential clinical trials conducted in the past decade have established the efficacy of pressure-based arterial disease assessment. In particular, strong evidence was produced that Fractional Flow Reserve (FFR)—an index based on invasive measurement of the pressure ratio distal and proximal to a given lesion—is highly effective in guiding the clinical decision to revascularise [1–5].

The pressure reduction taking place along a stenosis is nonlinear with respect to the stenotic severity, with a typical coronary artery exhibiting negligible pressure loss until a critical stenosis (∼ 50% diameter reduction) is reached, above which the loss begins to increase rapidly. A complex relationship between velocity and pressure drop is also observed, as flow regime transition will shift the balance of the dominant mechanisms underlying the energy loss. There have been many previous attempts to capture this multifactorial process with a simple, semi-

**Data Availability Statement:** All relevant data are within the paper and its Supporting information files.

**Funding:** This work was supported by Heart Research UK [RG2670/18/21]. This research was funded in whole, or in part, by the Wellcome Trust

[WT203148/Z/16/Z]. For the purpose of open access, the author has applied a CC BY public copyright licence to any Author Accepted Manuscript version arising from this submission.

empirical relationship based on the grounding theory, that does not feature many parameters nor invoke the need for a numerical solution [6–10]. In step with the clinical research, the interest in exploiting these reduced-order models for non-invasive clinical diagnosis too resurged, yet remains unfulfilled.

Such reduced-order models have the potential advantages over CFD of having a model substructure with easily identifiable roles thus highlighting the relevant haemodynamic mechanisms in individual lesions. Moreover, they require a fraction of the computational power owing to their algebraic construction. Though the accuracy of the existing reduced-order models have not reached a clinically satisfactory precision, as FFR is a threshold-based test with a binary decision outcome, reduced-order models have a strategic role in stratifying the clearcut cases from those that require a more detailed analysis, thus reducing the overall computational burden.

In order to improve the performance of the reduced-order models under disparate haemodynamic conditions, it is crucial that the model construction is underpinned by a sound theoretical basis. This is challenging as the flow through lesions operate near transitional turbulent regime and exhibits sensitive dependency to the lesion geometry. Capturing these relationships with the fewest empirical constants introduces a further challenge, as these constants do not generalise well across the different scales in physiological flows.

## Existing reduced-order models

Reduced-order models of stenotic vascular flow are functions of the flow velocity $\mathbf{u}(\mathbf{x}, t)$, which influences the components of energy loss along the stenosis. In practice, these models are more conveniently expressed in terms of the volume flow rate $Q(t)$, assuming incompressibility. Many models follow the general form

$$\Delta p = k_1 Q + k_2 Q^2 + k_3 \frac{\partial Q}{\partial t} \tag{1}$$

where $k_1$, $k_2$ and $k_3$ represent coefficients to be modelled, and $\Delta$p, the pressure drop. The conventional interpretation of Eq (1) is as a sum of viscous, turbulent, and inertial contributions, respectively. The first term addresses the effect of laminar flow through the stenosis, whereby pressure drop is linearly related to the flow rate via some modification of Poiseuille's law, taking into account the stenosis severity via $k_1$.

The second term of Eq (1) is characterised by a more ambiguous physical interpretation. On one hand, the quadratic velocity term bears superficial similarity to the Bernoulli priniciple in which the acceleration of blood through the stenosis leads to a pressure loss along a streamline, though this term is often attributed to the turbulence effects downstream of the stenosis and flow separation [11]. This formulation can be seen as early as 1963 [6] and has been adopted by many subsequent models [7, 8, 12, 13] without a rigorous definition of its physical mechanism. Its origin, however, is most likely to have been motivated by empirical considerations. Earlier studies used variations of Eq (1) to successfully reproduce experimental data in short axisymmetric and asymmetric stenoses [6, 14, 15].

Regardless, it is known that turbulence effects can contribute substantially to the pressure drop in arterial stenosis for both high and low Reynolds numbers $Re = \frac{\rho u D}{\mu}$; $u$ incoming velocity, $D$ unobstructed diameter, $\mu$ dynamic viscosity, $\rho$ density, [9] with the critical $Re$ changing with respect to the stenosis geometry to values even as low as $\approx 100$ [16]. Turbulent intensity might also increase due to the flow of red cells at $Re \approx 500$ or less, typically observed in flows through arterial stenoses [11, 17]. For values higher than the critical Reynolds number,

turbulence plays an increasing role in the haemodynamic parameters for both axisymmetric and eccentric stenoses [16, 18, 19].

Further developments of reduced-order models over the subsequent years have proposed similar relationships for pressure and flow rate, aimed at various specific applications [8, 13, 20, 21]. Studies comparing these models report contradicting results regarding their performance—some models that perform better than others with respect to reproducing 3D CFD results in one setting, may fail in other applications [22]. Of note, existing comparative studies are often limited to mild, sub-critical stenoses and a limited range of geometric variation [22–24]. Thus one of the objectives in the present study is to examine different lesion types, including eccentric stenoses. This is particularly important due to the high prevalence of eccentric lesions among patient populations. A study of plaque shapes using intracoronary ultrasound previously reported that 69% of lesions were eccentrically placed within the lumen [25]. Similarly, eccentric lesions accounted for 73% of 500 coronary lesions from a study of necropsy patients [26].

In the present study, we use 3D CFD simulations to quantify the shortcomings of the existing reduced-order models, and reexamine the derivation of turbulence contributions from first principle to yield an alternative formulation that better accounts for flow properties and the geometry of the stenosis. Our target is vessels of different sizes from larger to smaller arteries, and simulations employ three different stenosis types including axisymmetric and asymmetric, with a range of severities and unsteady inlet flow rates. The modified model is then compared with selected existing reduced-order models to quantify their accuracy against the CFD results.

## Materials and methods

### Reduced-order models for pressure drop

Here we consider two reduced-order models for benchmarking [8, 12]. Though neither of these models were derived for specific purposes to which they are applied here, both are used frequently in literature for predicting pressure in various types of stenotic vessels. The model of Young and Tsai [7, 12] is a trans-stenotic pressure drop model developed for a general vascular stenosis, with empirical parameters determined from a series of in vitro experiments. The model of Itu et al.[8] was developed for aortic coarctation employing a specific formulation for the viscous and the inertial terms, aimed at predicting pressure gradient in patient-specific stenosis geometries. These models treat the stenosis as one-dimensional fluid flow though an area-varying tube derived from empirical correlations.

**Young and Tsai model [7, 12–14].**   The pressure model proposed by Young and Tsai [12] is formulated as a quadratic polynomial of the flow rate, and is suitable for both steady and pulsatile flows. The model is expressed as [13]

$$\Delta p = \frac{4\mu K_v}{\pi D_0^3} Q + \frac{K_t \rho}{2 A_o^2} \left( \frac{A_o}{A_s} - 1 \right)^2 Q|Q| + \frac{\rho K_u L}{A_o} \frac{\partial Q}{\partial t} \tag{2}$$

Here $\rho$ is the blood density, $D_o$ the diameter of the unobstructed area of the vessel and $A_o$, $A_s$ are the unobstructed and stenosed areas of the vessel, respectively. $K_t$ and $K_u$, are empirical parameters obtained from experiments [12]. The values of $K_u$ and $K_t$ were empirically determined to be equal to 1.2 and 1.52 respectively. The parameter $K_v$ for the viscous loss was modelled as [14]

$$K_v = 32 \frac{L_s}{D_o} \left( \frac{A_o}{A_s} \right)^2 \tag{3}$$

where $L_s$ denotes the length of the stenosis and $L$ is a length calculated as in [7], which is also adopted here. The severity of stenosis is incorporated through the changing area ratio, which is present in both Eqs (2) and (3). The second term includes the energy dissipation due to what the authors attribute to turbulence, with $K_t$ the turbulence coefficient. The last term describes the inertial effects of the flow in a stenotic region. Cross-sectional areas of the stenosis are calculated from the corresponding radius, assuming that the stenosis has circular shape (although this might not be true). Based on experiments in blunt-ended plugs, Young [14] proposed the substitution of the length of the stenosis $L_s$ in Eq (3) in favour of $L_a$ where

$$L_a = 0.83L_s + 1.64D_s \qquad (4)$$

Here, $D_s$ refers to the diameter at the stenosis. The above expression is used for the present tests here for $K_v$. The model of Young and Tsai is applicable for time-dependent flows and a range of Reynolds numbers 100–5000. This model considered only short, smooth stenoses for different stenosis severities $50 - 90\%$. Subsequent studies suggest that the model can be used for calculating pressure drop for FFR, in mild degrees of occlusion [23] but will overestimate $\Delta p$ for higher stenosis severity [22].

**Itu model [8].**  The flow-dependent pressure-drop model proposed by Itu [8] is summarised by the following equations

$$\Delta p = K_v R_{vc} Q + \frac{\rho K_t}{2A_o^2}\left(\frac{A_o}{A_s} - 1\right)^2 Q|Q| + K_u L_u \frac{\partial Q}{\partial t} + K_c R_{vc} \bar{Q} \qquad (5)$$

$$K_v = 1 + 0.053\frac{A_s}{A_o}\alpha^2 \qquad (6)$$

$$R_{vc} = \frac{8\mu}{\pi}\int_0^{L_s}\frac{1}{R(x)^4}\,\mathrm{d}x, \qquad L_u = \frac{\rho}{\pi}\int_0^{L_s}\frac{1}{R(x)^2}\,\mathrm{d}x, \qquad \alpha = R_o\sqrt{\frac{\rho f}{\mu}} \qquad (7)$$

Similar to Young's model [14], the terms in Eq (5) represent the viscous, turbulent, and the inertial effects on pressure gradient respectively, with a final addition which is a continuous component to address the phase difference between the flow rate and the pressure drop. It is a function of the mean flow rate $\bar{Q}$ over one cardiac cycle [8]. The parameter $K_c = 0.0018\alpha^2$ is calculated from Womersley number, $\alpha$, which in turn is calculated from the patient heart rate ($HR$) and the angular frequency $\omega$, ($\omega = HR(2\pi/60)$).

The viscous loss term differs to the previous formulation in that it employs an integral evaluation of the viscous resistance ($R_{vc}$) to address the spatial variation in radius (Eq (7)). In addition, the viscous coefficient $K_v$ is modified to incorporate the Womersley number (Eq (6)). The coefficient $K_t$ for the turbulent term and $K_u$ remain the same as in Young and Tsai [12]. The formulations in Eq (7) were proposed for providing the model the capability to consider the shape of the coarctation [8]. For the inertia term, the length $L_u$ is added so that a more detailed description of the geometrical properties of the stenosis in the associated term in Eq (7) is included, adapting the model for patient-specific pressure predictions.

This model was developed for aortic coarctation with $Re$ up to 2000, and area stenosis severity ($1 - A_o/A_s$) between $40 - 60\%$, with a Womersley number $\alpha \approx 15$, but has been also capable of predicting pressure for coronary applications, of $Re$ approximately 100, with ($1 - A_o/A_s$) between $50 - 90\%$, with $\alpha \approx 1$ [22].

## Modified pressure-flow model

As mentioned above, increasingly, numerical studies have been employed to compare the reduced-order models against 3D simulations. But in some cases, the high accuracy reported in the original studies could not be reproduced in independent numerical investigations [22, 23]. This is likely due to the difference in the test scenarios which challenged the model beyond the original scope—vessel size, flow patterns and the severity of the stenosis are important determinants of the performance of these models that contain empirical correlations. No existing reduced-order model reproduces the CFD results in these broadly varying scenarios.

For this reason, we consider here an alternative formulation. For theoretical consistency across the whole formulation, we adopt and extend the approach of Ji et al. [27] which employed Lorentz's reciprocal theorem to decompose the fully-developed steady Navier-Stokes solution into contributing processes. Each term can then be treated to dimensional reduction, subject to appropriate assumptions.

In the following, we adopt the form of Lorentz reciprocal theorem previously presented [28, 29] to establish a relationship between the Stokes and Navier-Stokes stress field solutions within a common geometric domain [27]. Although this is an approximation, it is a useful one that, in particular, addresses the clinical threshold conditions of stenosed arteries under interest as will be demonstrated in our results. We note that similar approaches for modelling flows with moderate Re ranges have been successful previously [30].

For instance, the use of Stokes equation by including an auxiliary, nonphysical velocity field for pressure drop has been used in [31] and tested for Re = 560 and 765. The extension of Lorentz reciprocal theorem for obtaining an explicit relationship between Navier-Stokes equations and that of an auxiliary irrotational velocity field suitable for arbitrary Reynolds number flows is also possible [32].

As presented below, the use of the theorem will lead to an expression without the troublesome convective term and achieve a simpler breakdown of the mechanisms contributing to the pressure drop. Based on Brenner [28] the following expression links the two solutions

$$(\nabla \mathbf{u}) : \boldsymbol{\sigma}' = (\nabla \mathbf{u}') : \boldsymbol{\sigma} \tag{8}$$

where $\mathbf{u}$, $\boldsymbol{\sigma}$ are the velocity and stress fields satisfying Navier-Stokes, and likewise, $\mathbf{u}'$, $\boldsymbol{\sigma}'$, for Stokes flow. Integrating the above over the domain of volume $\Omega$ and surface $s$, and applying the no-slip condition at the lesion walls, yields an expression for the LHS of Eq (8) that contains the pressure drop in Stokes flow, $\Delta p'$, and the pressure drop in Navier-Stokes, $\Delta p$. The general integral expression of Eq (8) is

$$\Delta p = \Delta p' + \frac{2}{ReQ_m} (\int n_i u_j S'_{ij} \mathrm{d}s)_{in,out} - \frac{1}{Q_m} \left( \int (u_i u_j S'_{ij}) \mathrm{d}\Omega \right)$$
$$- \frac{2}{ReQ_m} (\int n_i u'_j S_{ij} \mathrm{d}s)_{in,out} + \frac{1}{Q_m} (\int n_i (u_i u_j u'_j) \mathrm{d}s)_{in,out} \tag{9}$$

Here $u_i$, $S_{ij}$ are the velocity components and strain rate tensor obtained from the Navier-Stokes equations respectively, and $u'_i$, $S'_{ij}$ the equivalent fluid properties obtained from Stokes flow. The flow rate obtained from Stokes flow is assumed to be equal to the flow rate from Navier-Stokes $Q_m$, thus simplifying the equation that connects $\Delta p$-$\Delta p'$ [27]. Eq (9) includes terms that can be grouped together into volume or surface integrals. This makes it explicit how pressure distribution in a stenosis is modulated by conditions at the inlet and outlet, and the flow inside the stenosis.

The simplifications applied hereafter depend on the flow characteristics and have to be carefully assessed to derive a simplified expression for calculating pressure. The first two surface integral terms in Eq (9) have been previously studied [27] for stenotic flows with a stenosis severity that varied within a range of 10% − 80% and were found to be small compared to the other terms in Eq (9) for a Reynolds number range of 1–1000. These terms are induced from the nonsymmetrical distribution of velocity at the inlet/outlet of stenosis. Since $S'_{ij}$ is the same in Stokes flow at both positions, the magnitude of the sum of these terms depended on the velocity difference which is small. Consequently, for higher $Re$, these terms also tend to be relatively small, since the integrals are multiplied by $1/Re \rightarrow 0$. Thus, these terms can be dropped from the pressure equation in the current analysis. Eq (9) is then simplified as

$$\Delta p = \Delta p' - \frac{1}{Q_m} \int (u_i u_j S'_{ij}) \mathrm{d}\Omega + \frac{1}{Q_m} \left( \int n_i (u_i u_j u'_j) \mathrm{d}s \right)_{in,out} \tag{10}$$

The first term of the RHS of Eq (10) is the pressure drop from Stokes flow and accounts for the contribution of the viscous effects to the pressure gradient at the stenosis. This term can be calculated by employing the lubrication theory for deriving a partial differential equation for the fluid pressure, p, with $\Delta p' = \Delta p'(Re, Q, A)$ where $A = A(x)$ is the profile along the flow direction $x$. If the geometry is axisymmetric and the profile of the vessel is known, then $\Delta p'$ can be easily calculated using the lubrication theory by integrating the radius of the vessel across the stenosis length obtaining the same expression as before [8, 27]. Nevertheless, we choose instead the simpler expression as used previously [13, 27] for reasons explained below. The first term is written for the volume flow rate, $Q$ as

$$\Delta p' = \frac{4K_v \mu}{\pi D^3} Q \tag{11}$$

where $K_v$ is an empirical coefficient as defined above [12, 14] in Eq (3).

The linear term here for viscous losses requires only the $A_o$, $A_s$ avoiding the use of the analytical expression for the stenosis profile [27]. Comparisons with expressions that employ the analytical stenosis profile shown in the results section illustrate that knowing the areas at these two positions provides a reasonable estimation for pressure for the pulsatile flows considered here, while offering simplicity.

The second term of the RHS in Eq (10) can be analysed by decomposing the velocity and strain rate tensor into a sum of time-averaged and fluctuating parts. As we detail in S1 Appendix, of the terms that arise with this derivation, it turns out the most significant contribution to the time-averaged part is given by the turbulence production related to the Reynolds stress [27]. In contrast, the remaining terms are identified to be negligible. Thus the term in the integral is approximated with the turbulent kinetic energy production which can be calculated from the dissipation rate of turbulent kinetic energy. For simplicity, it is assumed that $S'_{ij} \cong S_{ij}$ for this study. This is a reasonable estimation for steady-state flows (also shown in [27]) and for unsteady flows upstream of the stenosis, though is expected to be less accurate for pulsatile flows downstream of the stenosis. The second term of the RHS in Eq (10), becomes [33]

$$- \int (u_i u_j S'_{ij}) \mathrm{d}\Omega = \int_\Omega \epsilon \mathrm{d}\Omega \tag{12}$$

The aim is to simplify this integral so that it can be calculated without the need for the strain rate tensor which can be calculated only with three-dimesional CFD simulations. Since the pioneering work of Kolmogorov in the 1940s, much effort has been made to address approximate expressions for $\epsilon$ [34–40]. Often, the equilibrium dissipation law of the form $\epsilon = C_\epsilon u^3/\ell$ is used, where $C_\epsilon$ is treated as a constant, though there is an open debate as to whether $C_\epsilon$ is

dependent on $Re$. Previous experimental evidence suggest that $C_\epsilon$ tends to a constant value independent of Reynolds number when $Re$ is larger than 100 for tests involving turbulence generated by biplane square mesh grids [41], with values $\approx 1$. For homogeneous shear flows, investigations [42] concluded that $C_\epsilon$ is independent of $Re$ (when $Re > 100$) and that it is also weakly dependent on the shear. Here, we propose an empiricially-modelled expression that includes the impact of geometry and fluid properties together with this constant. This expression, as described next, adapts for different tests presented, addressing the dependency with respect to the flow regime for blood flows in complicated geometries. Using Eq (12) and approximating $\Omega = Q_m t$, yields the following

$$\frac{1}{Q_m} \int (u_i u_j S'_{ij}) d\Omega \cong \frac{1}{Q_m} \int_t C_\epsilon \frac{u^3}{\ell} \cdot d(Q_m t) = \frac{1}{Q_m} C_\epsilon \frac{u^3 Q_m}{\ell} \int_t dt = C_\epsilon t \frac{u^3}{\ell} \qquad (13)$$

A constant volume is considered to simplify and test here the above Eq (13) over a period $T = 1$ which is true only in a steady flow rate or for specific inlets for a pulsatile flow. Alternatively, a more accurate formulation can include the integral of the flow rate which can be easily calculated per cycle. Here, we simplify the expression for the numerical tests, and examine the modified pressure model with the new turbulence term. The length scale $\ell$ in the simplified turbulence term in Eq (13) can be taken as inlet diameter $D$. However, our experiments showed that a better fit for the integral length scale is the length $L_a$. We further account for possible geometrical contribution on this term by defining the constant $C_\epsilon$ as

$$C_\epsilon = \frac{2 C_m \nu \left(\frac{1}{A_s} - \frac{1}{A_0}\right)^2}{\rho \cdot S_D} \qquad (14)$$

where diameter stenosis severity $S_D = (1 - D_s/D_0)$ indicates the reduction in diameter for the stenosis and $C_m = 1 [kg^2\, s/m^4]$ is a constant parameter for dimensional consistency. The constant $C_\epsilon$ substitutes the empirical constant $K_t$ in the previous models for $\Delta p$. In this study, we use Eq (15) for modelling the contributions of turbulence in $\Delta p$ as

$$\frac{1}{Q_m} \int (u_i u_j S'_{ij}) d\Omega = C_\epsilon \frac{u^3}{L_a} \qquad (15)$$

For the remainder of the $\Delta p$ expression, the last term of the RHS of Eq (10) is also caused by the nonuniform distribution of velocity at the inlet and outlet of stenosis due to the presence of convective terms in the Navier-Stokes equations. Here, the contribution of this term is considered to be small compared to the other, Stokes flow term for turbulence, which is expected to be a reasonable assumption for sufficiently long lesions as shown by Ji et al. [27]. The wall shear stress is the highest at the lesion inlet where the boundary layer thickness becomes very small, and decreases gradually to the fully developed value. Therefore, pressure is higher at the entrance of the lesion, and the effect of the entrance region is always to increase the average friction factor for the entire lesion which may be significant for shorter segments. For long lesions this increase is expected to be less important. For a fully developed flow along a long lesion the integrals in this term will have the same magnitude at both inlet and outlet and will cancel out each other. Here, we choose not to consider the contribution of the surface integral of $n_i(u_i u_j u'_j)$ which is thus taken to be zero.

Since Eq (10) is for a fully developed steady flow through a stenosis, an extra term is added for accounting for the inertial effect of blood flow in the stenosis. This term is derived numerically from the rate of change of the volume intergral of the flux, which appears in the momentum equation [12] and is equal to $\frac{\partial}{\partial t} \int_\Omega \rho \mathbf{u} d\Omega$. Integrating over the length of stenosis, the latter

becomes $\rho L_s \frac{\partial Q}{\partial t}$, multiplied by $K_u$ as in the expression introduced in the model of Young and Tsai [12].

Substituting the inertia term, the linear term in Eqs (11) and (15) into Eq (10) the modified expression for pressure drop is obtained. Using the flow rate instead of velocity, our complete model for pressure drop can be written now in the following form

$$\Delta p = \frac{\mu K_v}{2\pi R_0^3} Q + \frac{C_\epsilon}{L_a A_o^3} Q^3 + \frac{\rho K_u L}{A_o} \frac{\partial Q}{\partial t} \tag{16}$$

## Computational fluid dynamics

CFD is widely regarded to be a reliable tool for the numerical investigation of applications related to physiological flows in stenosed arteries [43–46]. Here, the open-source code Open-FOAM [47] is used for the comparisons presented. OpenFOAM is a second-order finite-volume code where equations are solved on arbitrary polyhedral meshes in two or three-dimensions. It offers a fast development of user applications with a wide choice of discretization schemes, including different time, gradient, surface normal gradient schemes. The flows considered here are three-dimensional, unsteady and incompressible and the Newtonian assumption for blood viscosity is used. The mass and momentum equations are solved in time and space respectively in the context of an Eulerian grid

$$\frac{\partial u_j}{\partial x_j} = 0 \tag{17}$$

$$\rho \left[ \frac{\partial u_i}{\partial t} + u_j \frac{\partial u_i}{\partial x_j} \right] = -\frac{\partial p}{\partial x_i} + \frac{\partial \tau_{ij}}{\partial x_j} \tag{18}$$

where $u_i(x, t)$ represents the $i$-th component of the fluid velocity at a point in space, $x_i$, and time, $t$. Also $p(x, t)$ represents the static pressure, $\tau_{ij}(x, t)$, the viscous (or deviatoric) stresses, and $\rho$ the fluid density (instantaneous quantities are considered for now). The deviatoric stress tensor is given by $\tau_{ij} = 2\mu S_{ij} - \frac{2}{3}\mu\delta_{ij}S_{kk}$. From its definition $\tau_{kk} = 0$. The strain rate tensor is $S_{ij} = \frac{1}{2}\left(\frac{\partial u_i}{\partial x_j} + \frac{\partial u_j}{\partial x_i}\right)$, and $S_{kk} = \partial u_k/\partial x_k$. The dynamic viscosity of the fluid is $\mu$ and $\delta_{ij}$ denotes the Kronecker symbol. In case of incompressible flow, $S_{kk} = 0$ and the deviatoric stress tensor reduces to $\tau_{ij} = 2\mu S_{ij}$.

A linear interpolation scheme is used for the convective terms. The equations are solved implicitly using the Pressure implicit with splitting of operator, (PISO) algorithm [48] for coupling of pressure-velocity. For the tests here, two PISO iterations were employed, with each one solving the pressure equation. A first-order Euler scheme is used for time integration. Both laminar and Reynolds-Averaged Navier-Stokes(RANS) simulations are considered for the tests here. The $k - \omega$–SST model [49] is employed which has been proven the most suitable turbulence model among other RANS models for pipe-like flows. Previous studies suggest that both laminar and RANS may be appropriate for the range of $Re$ tested here [50, 51].

## Results

The numerical tests here included three different types of stenosis to allow a broader sampling of the geometrical influence on pressure drop. First two idealised axisymmetric stenosis

**Table 1. Characteristic lengths for the different geometries used for the tests (units in centimetres).**

|    | Geometry | $L_s$ | D |
|----|----------|-------|---|
| G1 | Large axisymmetric short | 10 | 5.08 |
| G2 | Small axisymmetric short | 1 | 0.508 |
| G3 | Large Axisymmetric long | 40 | 5.08 |
| G4 | Eccentric | 1.4 | 0.4 |

$L_s$ is the stenosis length. $D$ is the diameter of the lesion.

geometries (denoted as G1 and G2) and a long axisymmetric stenosis (denoted as G3) were used. Eccentric stenoses (denoted by G4) were also considered in the present study.

The length of the stenosis $L_s$ and the unconstricted diameter of stenosis artery are shown in Table 1. They range from scales representative of aorta down to proximal coronary vessels. The domain considered in the CFD simulations included a distance of four unconstricted diameters upstream of the stenosis which is considered in the pressure-drop calculations.

The stenosis severity $S_D$ varied in these geometries from 50% to 90%. Both laminar and RANS simulations were performed for all cases for comparison. Two types of boundary conditions were used for the inlet velocity for the pulsating unsteady blood flow. BC1 is of the form $u(t) = u_0 + A\sin(2\pi f t)$, where $A, f, u_0$ are the amplitude, frequency and the offset level for the sinusoidal flow, which for the first idealised stenosis in Fig 2 and the long stenosis Fig 4 were $A = 0.01 m/s, f = 1 s^{-1}, u_0 = 0.02 m/s$. BC2 is a pulsatile periodic flow waveform with a period of $T = 1s$ (see Fig 1). For the second axisymmetric and eccentric geometries tests in Figs 3 and 5, for the inlet velocity BC1, these were $A = 0.1, f = 1, u_0 = 0.2$ and for BC2 the velocity was ten times the one used in the larger idealised stenosis tests while preserving similar values for inlet $Re$ for both series of tests up to 500.

In both upstream conditions BC1, BC2, the flow was assumed to be fully developed downstream and a Neumann boundary condition was used at the outlet (zero gradient). Hence, a set of 54 CFD simulations were performed in total. The CFD results were compared with the results obtained from the three reduced-order models described in the Methodology section.

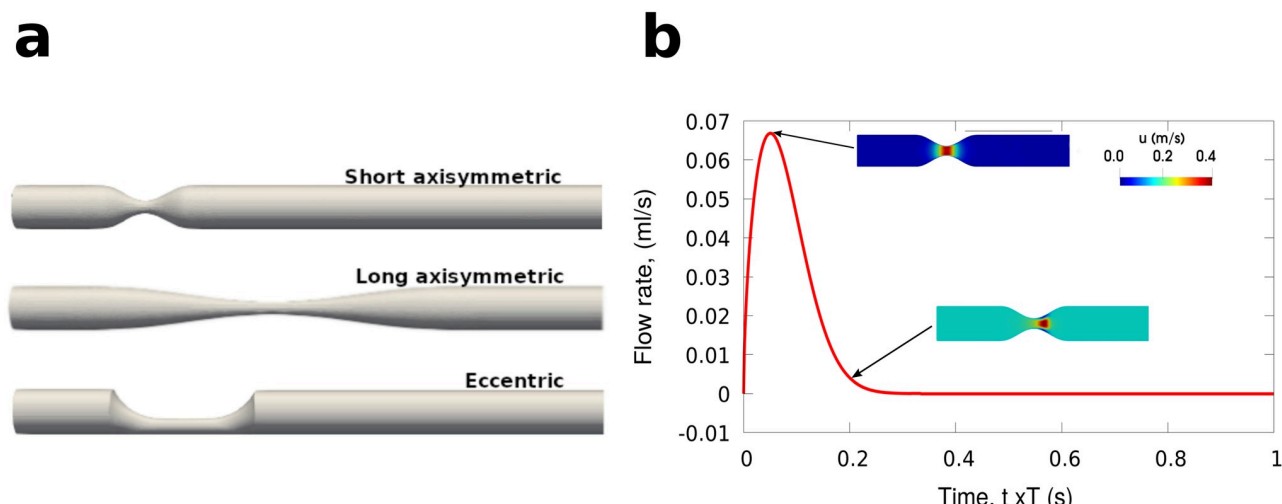

**Fig 1. Types of stenosis and boundary condition BC2.** Illustration of geometries used for the model testing and CFD simulations (not to scale) and the inlet boundary condition for BC2. a: types of stenosis geometries used. b: inlet boundary condition for BC2.

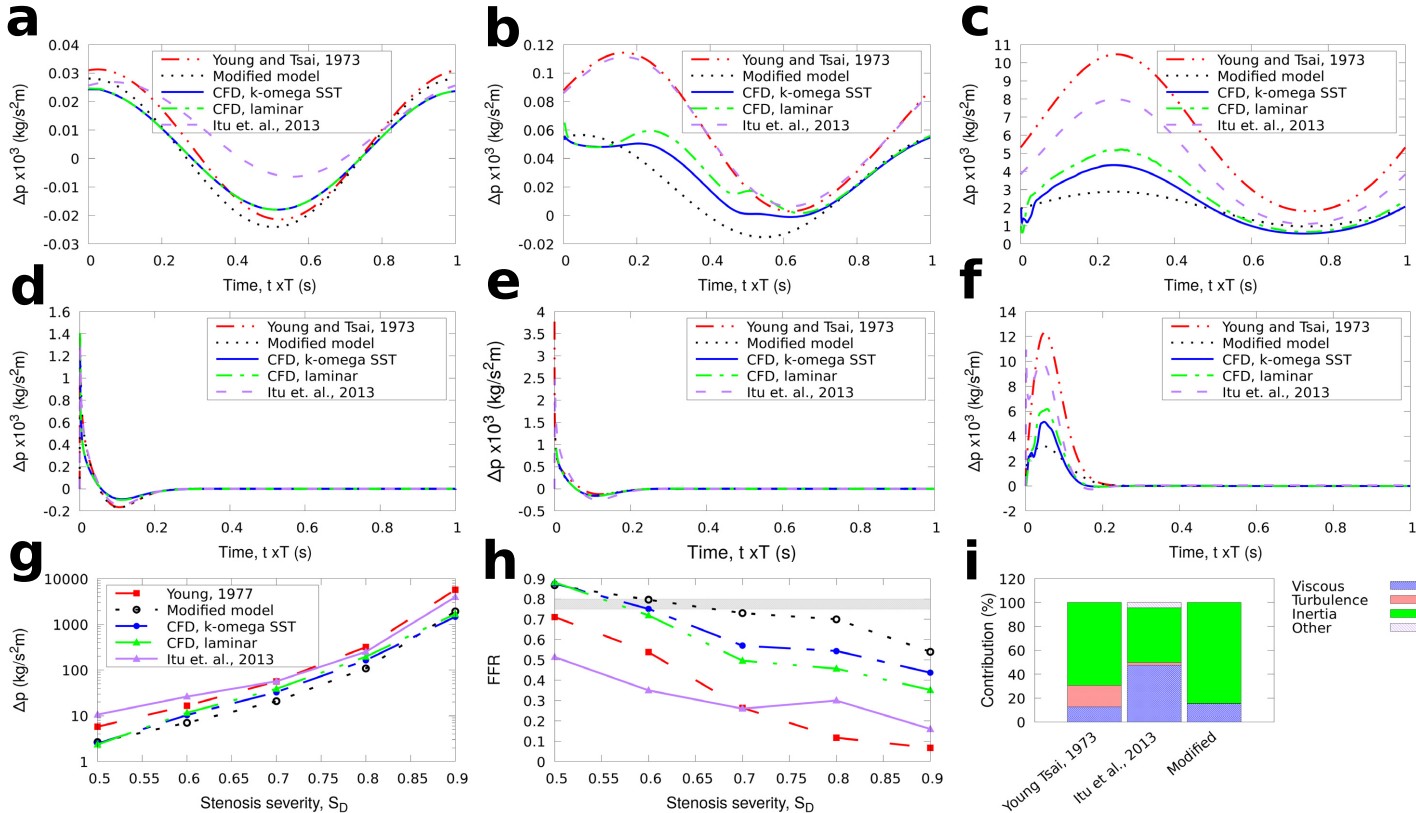

**Fig 2. G1 stenosis.** Comparison of results for the reduced-order models and full 3D CFD simulations for the large axisymmetric stenosis (G1) with 50%, 70%, and 90% stenosis severity. The average pressure for one cycle is also shown for BC1. The FFR is calculated as $1 - \bar{\Delta p}/\bar{p}_o$ with the diagnostically significant range ($0.75 \leq FFR \leq 0.8$) indicated by the grey stripe. a: types of stenosis geometries used. b: inlet boundary condition for BC2. a: BC1, stenosis severity 50%, b: BC1, stenosis severity 70%, c: BC1, stenosis severity 90%, d: BC2, stenosis severity 50%, e: BC2, stenosis severity 70%, f: BC2, stenosis severity 90%, g: BC1, averaged pressure, h: BC1, FFR, i: Contribution to $\Delta p$, $S_D = 0.6$.

The value of FFR, measured in reference to aortic root pressure $p_o$, can be approximated as the amount of pressure drop across a stenosis based on the avaraged values for $\Delta p(\text{FFR}1 - \bar{\Delta p}/\bar{p}_o)$. The results for $\Delta p$ and FFR for $S_D = 50 - 90\%$ are shown first for the two idealised, axisymmetric stenoses in Figs 2 and 3. Next, the results for the pressure drop and FFR through time and the time-averaged values for each stenosis severity for BC1 and BC2 are shown for the long axisymmetric and eccentric stenoses in Figs 4 and 5 respectively. In all cases the percentage contribution of each term in the total calculated $\Delta p$ is shown for the average velocity used in BC1 and $S_D = 0.6$.

## Discussion

### Short axisymmetric lesions (G1, G2)

The results for the idealised, axisymmetric stenoses with $L_s/D = 2$ in Fig 2 for the large diameter case (G1), and Fig 3 for the small diameter case (G2) illustrate that the proposed model produced results that better approximated the CFD solutions for both inlet velocity boundary conditions compared to the existing reduced-order models. For all stenosis severity levels, the calculated $\Delta p$ using Eq (16) was smaller than the other models. The results for the time-

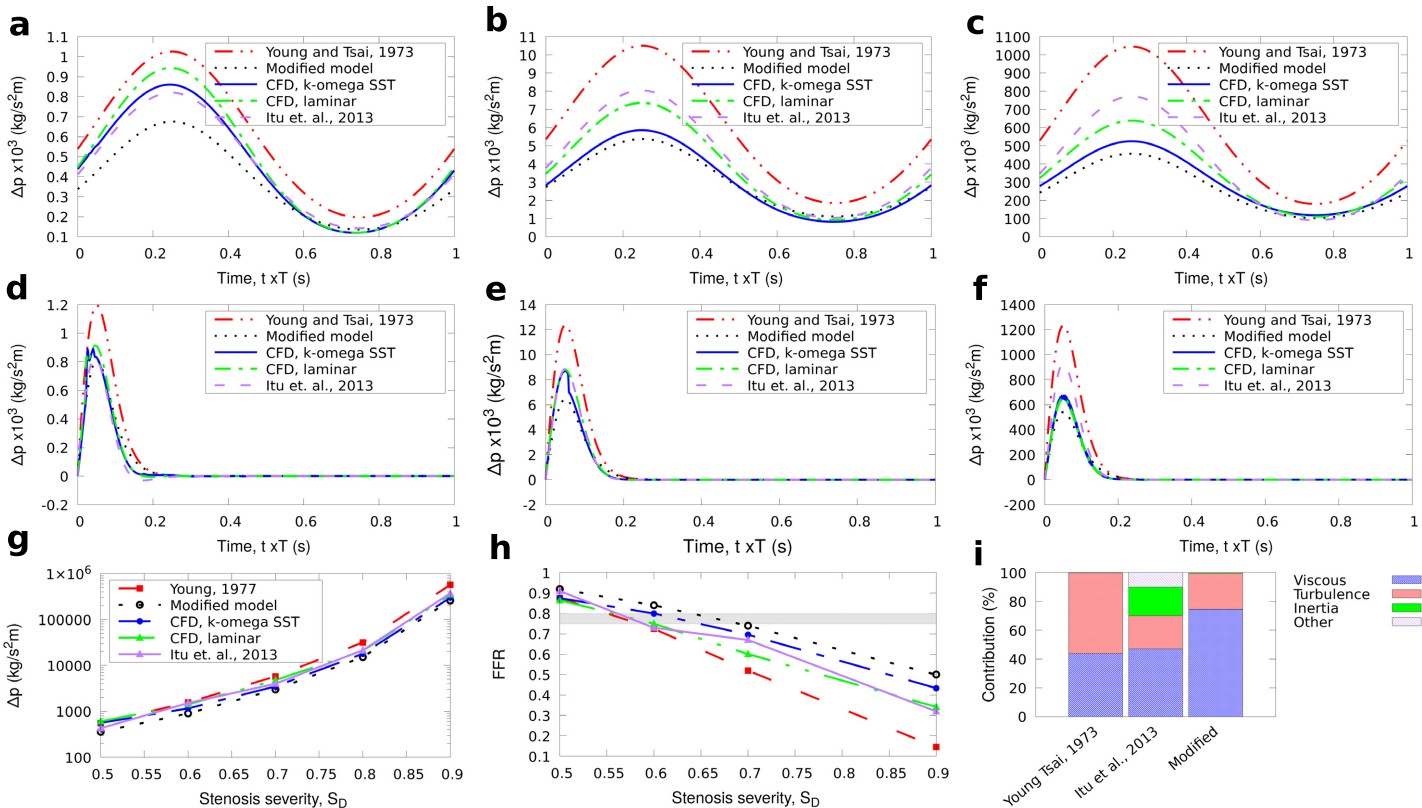

**Fig 3. G2 stenosis.** Comparison of results for the reduced-order models and full 3D CFD simulations for the small axisymmetric stenosis (G2) with 50%, 70%, and 90% stenosis severity. The average pressure for one cycle is also shown for BC1. The FFR is calculated as $1 - \overline{\Delta p}/\overline{p_o}$ with the diagnostically significant range ($0.75 \leq FFR \leq 0.8$) indicated by the grey stripe. a: types of stenosis geometries used. b: inlet boundary condition for BC2. a: BC1, stenosis severity 50%, b: BC1, stenosis severity 70%, c: BC1, stenosis severity 90%, d: BC2, stenosis severity 50%, e: BC2, stenosis severity 70%, f: BC2, stenosis severity 90%, g: BC1, averaged pressure, h: BC1, FFR, i: Contribution to $\Delta p$, $S_D = 0.6$.

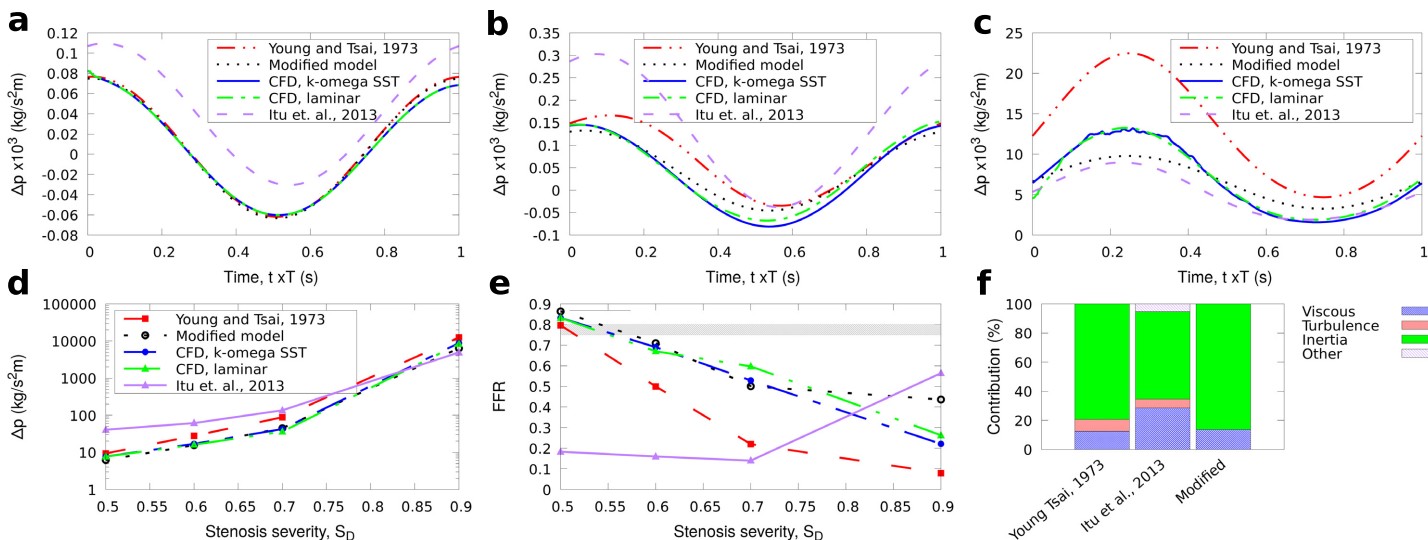

**Fig 4. G3 stenosis.** Comparison of results for the reduced-order models and full 3D CFD simulations for the long axisymmetric stenosis (G3) with 50%, 70%, and 90% stenosis severity. The average pressure and FFR for one cycle are also shown for BC1 with the diagnostically significant area marked with grey stripes. a: BC1, stenosis severity 50%, b: BC1, stenosis severity 70%, c: BC1, stenosis severity 90%, d: BC1, averaged pressure, e: BC1, FFR, f: Contribution to $\Delta p$, $S_D = 0.6$.

averaged pressure during the first three cycles (Figs 2g and 3g) are also close to the 3D CFD results when using the proposed expression for $\Delta p$, with 3–30% deviation from the RANS results.

The modified term for turbulence is generally smaller in magnitude than in the model of Young [12] and Itu [8]. In their experiments, Young et al. [14] illustrated that localised turbulence or flow separation begins to occur at low $Re \approx 200$ or more for stenotic flows through femoral arteries of anaesthetised dogs. Young and Tsai model uses the parameter $K_t$ that remains constant for all geometries tested, being consistent with previous work [14]. The second term in Eq (16) is $\approx K u^3$, where $K = 0.1, 1, 10$ for $S_D = 0.5, 0.7, 0.9$ respectively for the larger idealised geometry tests (G1) in Fig 2 which resulted in a very small contribution to the total $\Delta p$ as shown in Fig 2i. In the smaller axisymmetric geometries tests (G2) in which the velocity at the end of the stenosis is higher than in the larger axisymmetric tests with higher local $Re$, the magnitude of the turbulence term is much higher resulting in a more significant contribution to $\Delta p$ as shown in Fig 3i. The term accounting for the turbulence effects in the other reduced-order models tended to over-predict this term for both BC1 and BC2. Our formulation is shown to adjust to the new stenosis severity thus avoiding the use of extra tuning of parameters.

Of the three terms in the pressure drop equation in the lower stenosis severity cases, the viscous and turbulence terms are smaller than the inertia term, which has the highest contribution to the total pressure drop. For higher stenosis severities, the viscous term becomes increasingly important. The magnitude of the turbulence term also rises which is expected since the local $Re$ increases significantly for $S_D > 0.7$. Using a smaller value of $K_t = 1$ for the models of Young and Tsai, and Itu et al., as originally proposed [12] leads to a lower $\Delta p$ that is approximately 1.5 times smaller, but even with this modification, the pressure drop still remains significantly larger than those obtained with our proposed model.

Although the values of the inertial and viscous terms in Eq (5) take into account the specific shape of the stenosis, the results of Eq (5) were not improved since the contribution of the turbulence term (which is the same as in Eq (2)) was significantly higher than in the modified model. The continuous term in Eq (5) has a small impact on the pressure drop for the test here. For instance, for the large axisymmetric short stenosis it always remained less than 8% than the total $\Delta p$. Their large differences with 3D CFD may be explained considering that the model of Itu et al.[8] was developed in the context of aortic coarctation for Reynolds numbers of the order of 2000, as well as relatively low stenosis severities of area reduction in the range 40% to 60%, whereas in our tests, $Re$ varied from 100 to 500 and the stenosis considered had radius reduction that varied from 50% up to 90%, reflecting the more typical conditions in the coronary artery.

### Long axisymmetric lesion (G3)

Since all terms in the reduced-order pressure drop models strongly depend on the geometry and type of lesion, we further employed longer stenoses keeping the same diameter $D_o$ and boundary conditions as before. The effect of the stenosis length on pressure was examined using the idealised axisymmetric lesion G1 extended by 400%. The new set of geometries had $L_s/D = 8$ and are denoted with G3. The results for Eq (16) for each stenosis severity with BC1 shown in Fig 4 were close to the 3D CFD results when $S_D \leq 0.7$ with error increasing at larger $S_D$, although $\Delta p$ remained closer to RANS than the other reduced-order models with a maximum deviation with the RANS results approximately 6%.

The predictions of the averaged $\Delta p$ in Fig 4d and 4e are closer to RANS than the ones in the short stenosis cases Figs 2g, 2h, 3g and 3h. While the Reynolds number for the same $S_D$

remained generally the same for the simulations in Figs 2–4, the profile of the velocity was generally different at the end of the stenosis: for the long stenosis the velocity at the flow direction is parabolic downstream the end of stenosis, whereas in the short stenosis cases a jet-like regime occurs which has higher velocity at the jet core which occupied less than 50% of the stenosis diameter. Since the velocity is uniform at the inlet and the outlet of the stenosis, the integrals of $u_i u_j u_j'$ have approximately the same value and their sum is assumed to be approximately zero. Consequently, the assumption for zero contribution of the last term in Eq (10) is valid for the long stenosis tests with $L_s/D = 8$ which is also illustrated from the comparisons with the short stenosis results with $L_s/D = 2$.

Although, it has been observed experimentally that the ratio $L_s/D$ may have limited effect for severely constricted lesions with the same stenosis severity for steady state flows [7], for the long stenosis tests, the calculated $\Delta p$ differed from those obtained with the corresponding short lesions, and were higher for all models and CFD simulations (usually three times or more for the 3-cycle average). The laminar and RANS results were close for the long geometries used here, which was only observed for milder $S_D$ in the short axisymmetric stenoses tests. When increasing the stenosis length, the characteristic Reynolds number of the initial local turbulence (observed flow instabilities) too increases, in line with previous experimental studies [16].

The inertia term had the most significant contribution for all the reduced-order models with the contribution of the turbulence term being small compared to viscous and inertia terms for $S_D = 0.6$ Fig 4f. It was observed that the relative contribution of the various terms in the different pressure drop equations can differ significantly. For instance, the viscous term is not significant in lower stenosis severities but became the highest for $S_D = 0.9$ for the tested long axisymmetric geometries. Apart from the geometry, this contribution very likely depends on the temporal flow characteristics of the problem since it can also cancel out when averaging over a cycle. Nevertheless, results for BC1 in Fig 4d did not differ qualitatively as in the short axisymmetric tests, revealing similar trends of the Eq (16) with respect to $S_D$ which is also revealed in Fig 4e for the calculated FFR.

## Eccentric lesion (G4)

This set of geometries had $L_s/D = 3.5$. As with above, the results of the proposed formulation were closer to the RANS results compared to the other reduced-order models for pressure drop. The contribution of the cubic turbulence term in Eq (16) was significant and accounted for $25 - 50\%$ of the calculated pressure drop for the different boundary conditions (Fig 5i).

It is seen that laminar simulations can be adequate in certain stenosis conditions (Fig 5a and 5d). Increasing the stenosis severity increases the chances for turbulence developing distal to the stenosis at low Reynolds numbers. For instance, transition is most likely to occur at a Re $\approx$ 200–300, for stenosis severities more than 90% [52]. Turbulence effects are more prevalent in the diverging section of the stenosis than the converging part, with a significant impact on both pressure and wall shear stress. Interestingly, the proposed expression for the turbulence term in Eq (16) is capable of predicting different geometries with both short and long stenosis lengths as shown in the results. In the model of Young and Tsai, the empirical parameters used in the turbulence term appear to cause differences with CFD results, overestimating pressure drop as compared to the modified model, which uses the same viscous losses and inertia terms. On the other hand, the model of Itu et al. demonstrated an improved behaviour compared to Young and Tsai model. The viscous losses and the turbulence losses terms both usually constituted the terms that were primarily responsible for the values of $\Delta p$ for the tests in

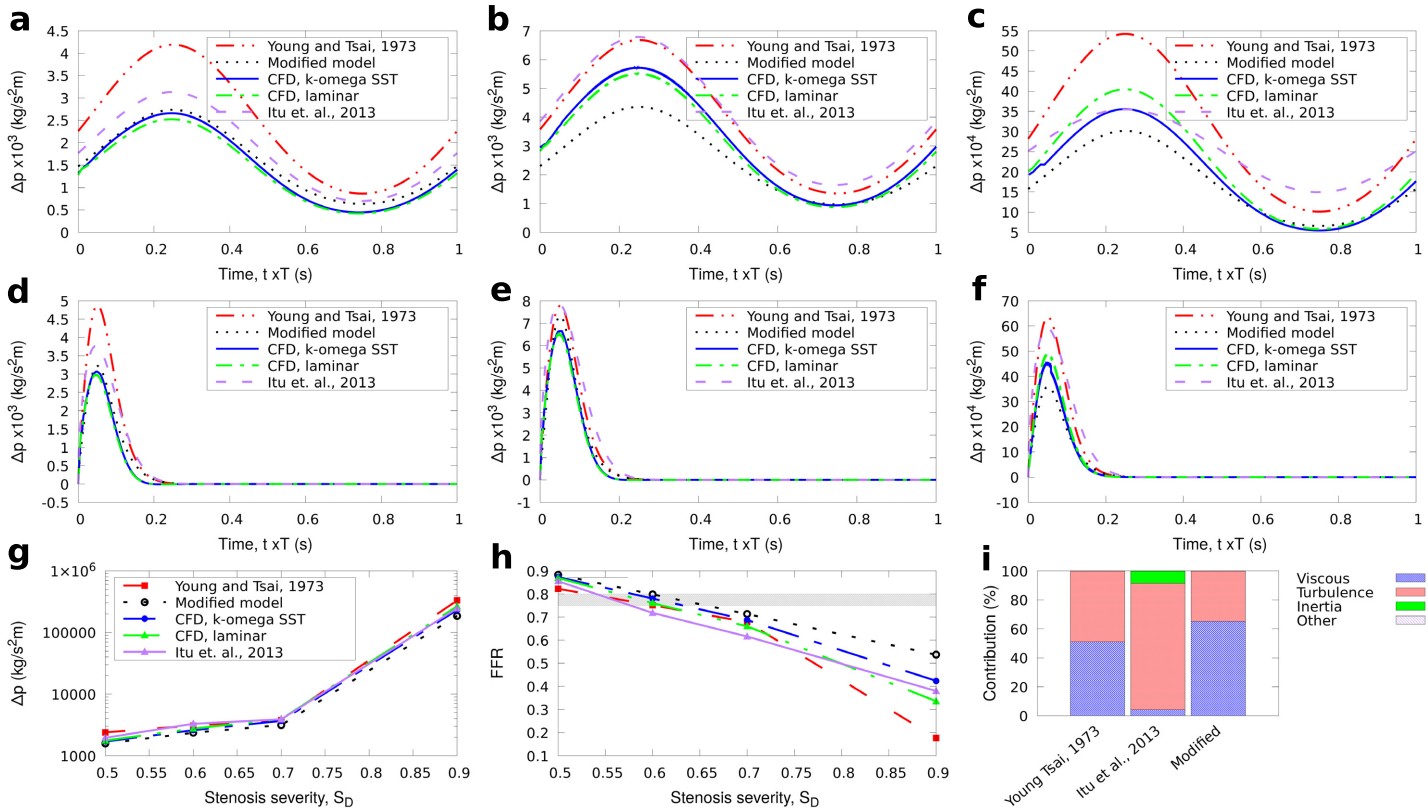

**Fig 5. G4 stenosis.** Comparison of results for the reduced-order models and full 3D CFD simulations for the eccentric stenosis (G4) with 50%, 70%, and 90% stenosis severity. The average pressure and FFR for one cycle are also shown for BC1 with the diagnostically significant area marked with grey stripes. a: types of stenosis geometries used. b: inlet boundary condition for BC2. a: BC1, stenosis severity 50%, b: BC1, stenosis severity 70%, c: BC1, stenosis severity 90%, d: BC2, stenosis severity 50%, e: BC2, stenosis severity 70%, f: BC2, stenosis severity 90%, g: BC1, averaged pressure, h: BC1, FFR, i: Contribution to $\Delta p$, $S_D$ = 0.6.

Fig 5. Whereas, according to previous reports, the effect due to inertial processes might be less important for arterial flows through stenoses with severity higher than 50%[11].

Particularly with regard to FFR calculation, a steady flow approximation can instead be used provided that the mean values for a complete cycle are employed during a pulsatile flow. In this case, Eq (16) becomes of the form: $\Delta p = AQ + BQ^3$. Similarly, in several previous studies [6, 14, 15, 53] expressions with two coefficients are used for the steady problem. The results shown in Figs 5a–5c and 5g and 5h reveal that these two terms are adequate here for obtaining results close to the 3D CFD results, providing an FFR that remained reasonably close to CFD within the diagnostically sensitive range.

Interestingly, from the tests here, the balance of the associated terms on the total pressure-drop calculation changes with respect to $S_D$. For instance for the eccentric stenosis tests, the viscous losses term in Eq (16), was, on average, three times more than the turbulence term (second term in Eq (16)) for $S_D$ = 0.5 and 1.35 times more for $S_D$ = 0.7. On the contrary, the turbulence term was six times higher than the viscous losses term for $S_D$ = 0.9 (Fig 5g). As shown in the comparisons of the calculated pressure drop across the respective stenosis from the reduced-order models and the CFD simulations in Figs 2, 4 and 5, the new modification for the turbulence term using Eq (14) can provide predictions for the time evolution of the pressure change in a stenosis close to the CFD simulations. Consequently, despite the different contributions of the involved terms, their sum remained adequately close to the 3D CFD

results for both BC1 and BC2 which is more clearly illustrated in Figs 2h, 3h, 4e and 5h for the calculated FFR.

## FFR evaluation

From a clinical translational perspective, FFR can be classified into three categories—negative ($\geq 0.8$), positive ($\leq 0.75$) and intermediate ($0.75 - 0.8$). In the intermediate range, increased precision is demanded of any numerical scheme in order to minimise categorical diagnostic error. While our predictions of FFR produced increasing quantitative error in higher severities, all predictions remained correct in their classifications according to the RANS results, in every simulation performed. This classification error of 0% was produced by no other reduced-order models.

The intermediate FFR range tended to correspond to a milder $S_D$ of around 60%, where the contribution of the new turbulence term to $\Delta p$ was around $0 - 35\%$. Towards higher $S_D$ of around 90% our FFR prediction suffers from substantial overestimation, however, in these very severe cases FFR would add little to the clinical management as the decision to revascularise would rest on other factors e.g. procedural safety. Most interventional laboratories in practice would only perform an invasive FFR if the angiographic severity was ambiguous (intermediate).

Our model performed least accurately for FFR prediction in the short lesions with $L_s/D = 2$ but gave predictions with 13% maximum difference with the RANS results for every long axisymmetric and eccentric tests for $S_D \leq 0.6$ which was lower than the other models. In the case of the short stenoses, the surface integrals of $u_i u_j u_j'$ at the two ends of the stenosis which are induced when integrating the term $(\nabla \mathbf{u}'): \boldsymbol{\sigma}$ in Eq (8) is expected to be non-zero. For flows that are not fully developed such as the ones here with $L_s/D = 2$, the flow is nonuniform at the outlet of the stenosis. Including these terms in the pressure drop calculations for these cases and for $L_s/D \leq 2$ could further improve the performance of the model. On the other hand, for $L_s/D \geq 3.5$ the results indicate that the net sum of these terms can be assumed to have insignificant contribution, since the flow was uniform at the end of stenosis.

Finally, in our results a steady state simplification of Eq (16) produced an accurate approximation of the time-averaged dynamic formulation. The elimination of an empirical constant $K_u$ is an added advantage. Nevertheless, these results may have been favoured by the boundary conditions we chose which were approximately "symmetric" in time, such that $\frac{\partial Q}{\partial t}$ will average to be near zero over a cycle. Further investigation should be performed using patient-specific measurements to verify that this assumption is reasonable in clinical application.

## Model evaluation

So far, we have addressed the physical ambiguity of (what is usually) the quadratic term, by explicitly modelling it using the assumption of turbulent energy dissipation. Our results and discussion above show that the proposed model improves the performance over previous ROMs. Furthermore, by considering the contribution of the different terms, it is evident that the improvement was due to the behaviour of the new cubic turbulence term (i.e. it does not overestimate its own effects).

However, these conclusions are conditional upon our own CFD results being accurate. In order to challenge this assumption, we adopted the model comparison outlined in a recent review [54] as a benchmark. The study compared 0D (Young and Tsai model), 1D (standard formulation) and 2D (Multiring [55]), and *in vivo* arterial measurements [56]. Using the simulation set up described in the paper, we added the results from our CFD, the proposed ROM, and the model of Itu et al. as reference.

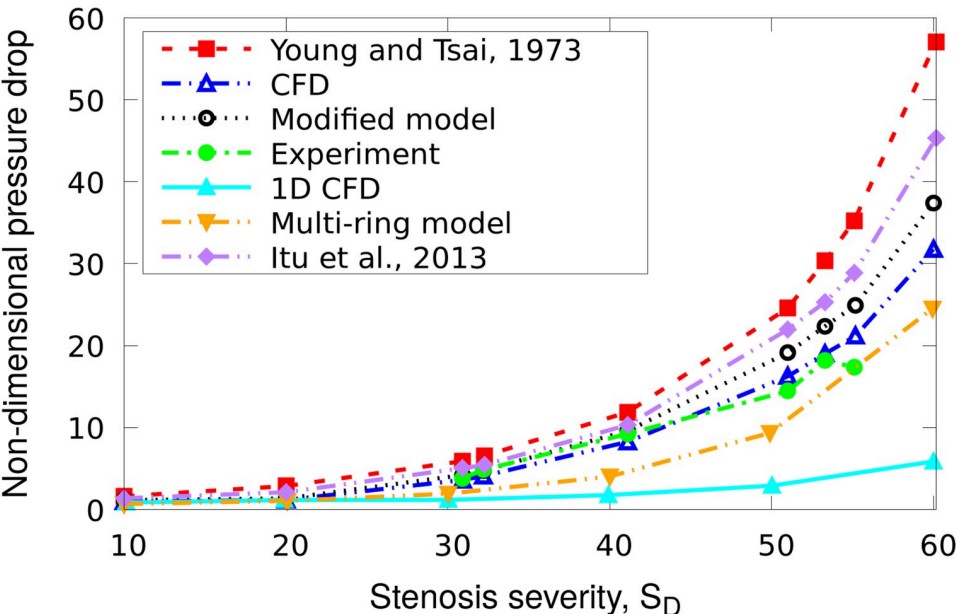

**Fig 6. Model comparisons.** Comparison of dimensionless pressure drops predicted by models of varying dimensionalities in an axisymmetrically stenosed tube, adopted from [54]. Young and Tsai 0D model, 1D model and 2D multi-ring model are compared to the in vivo measurements [56] as in the original publication, to which we have added our 3D CFD, the proposed ROM and model of Itu et al.

Fig 6 shows that 3D CFD results are the most accurate with respect to the *in vivo* data up to the final data point. It is less accurate than the 2D model at the final data point, however, the fact that the pressure drop backed down from a less severe stenosis suggests there may be an experimental issue. The results of our ROM is consistent with our other test cases in that its prediction of $\Delta p$ was more accurate than [8], which in turn was more accurate than [12]. Interestingly, the accuracy of the proposed ROM in predicting $\Delta p$ was roughly equivalent to the 2D model at higher levels of severity and superior in mild stenoses.

## Modelling approaches

The 0D algebraic modelling is not the only approach to reduced-order modelling, a term which includes any model that simplifies the full CFD. There have been recent studies assessing the suitability of the 1D and 2D models as well as other novel schemes (based on e.g. projecting the full Navier-Stokes solution to a lower-dimensional subspace using orthogonal decomposition [57], combining 0D ROM with machine learning [58]). However, the consensus emerging in the wider community appears to be that a well-designed 0D ROM is preferred due to its simplicity, lighter computational footprint, and in most cases, accuracy that is bested only by 3D CFD [22, 23, 54].

In context of patient-specific coronary simulations, 0D ROMs have been routinely combined with higher-dimensional models to compute network flows. Indeed, this multiscale modelling strategy has become standard in patient-specific FFR studies where 0D components have been particularly effective at addressing the stenosis and distal boundary conditions. Interestingly, the recent literature is split into two different camps depending on whether CCTA or angiography provides the imaging data, each with distinct norms and philosophies. CCTA-based FFR studies tend to employ a network model and focus on novel methodological

applications [57, 59–61]. Angiography-based virtual FFR studies focus on less extensive modelling domain and methods, favouring single vessel models and embracing simplifications that are practically useful. Despite this, most recent vFFR work featured prospective multicentre trials, yielding diagnostic accuracy in the region of 90% or more [62, 63]. These different trajectories are driven by the information extractable from the respective imaging modalities (vascular anatomy and auxiliary features that help to tune distal/proximal conditions), and the timescale of the application ("on table" vs offline). Our present work being but one component in the multiscale framework, we have evaluated it on the basis of its own merit here—and while the results presented verify its efficacy in a localised stenosis setting, further modelling and validation should be performed in the application of intended purpose.

Lastly, there have also been studies aimed at using machine learning techniques to characterise physics of fluid, including the hemodynamics of stenosed vessels. While this approach may be superior at identifying nonlinear complexities than the hand-tuned models considered here, the interpretable model structure of the latter is beneficial to gaining clinician's trust, where machine learning is largely a black box and may produce spurious results outside the training range. In addition, the mechanistic modelling approach used here requires far fewer CFD simulations to develop and tune. It is, however, possible to combine the two approaches for further benefit. This is an ongoing avenue we are pursuing and thus we consign this discussion to outside of the present scope.

### Physiological and clinical relevance

Our model is designed specifically to address the local pressure drop in the regions adjacent a stenosis. As such, many physiological details that are otherwise necessary to capture the hemodynamics of a network or a whole segment become unnecessary. For instance, elastance of the vessel walls which modulates pulse waves can be safely disregarded for pressure estimation in the vicinity of an atheroma which tends to be stiffer and calcified. Supporting our use of a rigid wall model further, is that the presence of atherosclerosis is associated with elevated general wall stiffness throughout the vascular network [64]. Simulation studies which have employed elastic walled flow models consistently report a limited role of the vascular elastance on pressure drop (2D [54]; 3D FSI [22]).

The diameters of the test geometries in this work range between $4mm$ to $5cm$, and were chosen to represent typical calibers of human aorta (G1/G3), common carotid (G2) and coronary arteries (G4). In terms of the coronary vessels, the chosen diameters are well within those found in the first or second generation vessels as measured by CCTA, such as Left Main ($3.5 \pm 0.8mm$), LAD ($3.2 \pm 0.7mm$), LCX ($3.0 \pm 0.7mm$) and RCA ($3.4 \pm 0.6mm$) [65]. Most hemodynamically consequential lesions occur within these large coronary segments, due to their larger distended tissue volume.

Finally, there have been numerous alternative physiological indices proposed in the recent years for interventional decision making; in the coronary circulation, these include iFR, QFR, and a variety of non-hyperemic pressure ratios (RFR, DFR, dPR, resting pd/pa) offered by various vendors [66]. Nevertheless, we focus our attention on FFR here since few are supported at current time by large prospective clinical evidence based on the patient outcome as FFR has been. Those which have undergone large-scale clinical investigation are often compared to FFR as the gold standard, and for the foreseeable future, will remain so.

### Limitations and future work

Many idealisations have been made in the geometry and simulation parameters of the model evaluation. Real atherosclerotic lesions tend to feature an overwhelmingly eccentric lumen

that is irregular in shape [26]. In addition, although the present testing set up has produced flow regimes with varying contributions from viscous, inertia and turbulence terms, it lacked the complexity found in the physiological settings. Future studies should employ these refinements to quantify how much the model performance would deteriorate, which will help to identify new targets for model improvement.

In addition, sensitivity analyses from both CCTA [61, 67] and angiography-based [68] patient-specific FFR modelling studies agree that estimation of boundary condition is the dominant factor contributing to the overall error, about twice as more significant than the error in vascular modelling itself. Future work should therefore involve embedding the proposed model within patient-specific settings and assess its performance across a range of physiological conditions against measured FFRs.

## Conclusion

Combining a CFD-based comparison with a variety of stenosis types and severities, in this work, we produced new data on the efficacy of existing reduced-order models for predicting FFR. Following on, we reexamined the theoretical basis of these models, and addressed their empirical shortcomings via a novel approach utilising Lorentz's theorem. Our analysis revealed the ad hoc formulations of past models to be suboptimal; using a model of turbulent energy dissipation, an improved model was recovered. The resulting expression is still simple, but produced superior pressure drop prediction in both aorta and coronary-sized vessels as verified by numerical testing. While the FFR predictions of the proposed model did not reach a clinical-level precision ($|\Delta| < 0.01$), it produced zero classification error. Further investigation is under way to quantify the model performance in the intermediate lesions in a more extensive setting.

## Supporting information

**S1 Appendix. Derivation of turbulence term.** Explanation of the second term derivation in the reduced order model.
(PDF)

## Author Contributions

**Conceptualization:** Konstantinos G. Lyras, Jack Lee.

**Formal analysis:** Konstantinos G. Lyras, Jack Lee.

**Investigation:** Konstantinos G. Lyras, Jack Lee.

**Methodology:** Konstantinos G. Lyras, Jack Lee.

**Validation:** Konstantinos G. Lyras, Jack Lee.

**Visualization:** Konstantinos G. Lyras, Jack Lee.

**Writing – original draft:** Konstantinos G. Lyras, Jack Lee.

**Writing – review & editing:** Konstantinos G. Lyras, Jack Lee.

**funding-acquisition:** Jack Lee.

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
