## [Decision Letter · Decision Letter 0]

27 Apr 2021

PONE-D-21-04950

Numerical investigation of reduced-order modelling in arterial stenosis

PLOS ONE

Dear Dr. Lyras,

Thank you for submitting your manuscript to PLOS ONE. After careful consideration, we feel that it has merit but does not fully meet PLOS ONE’s publication criteria as it currently stands. Therefore, we invite you to submit a revised version of the manuscript that addresses the points raised during the review process.

We look forward to receiving your revised manuscript.

Kind regards,

Fang-Bao Tian

Academic Editor

PLOS ONE

Additional Editor Comments:

Thank you for submitting your work to the Plos One. Two independent reviewers have commented on it. One criticizes the over-simplified model which seems far away from the realistic condition. I do agree with this reviewer from the clinical perspective. That said, this work adds values to the engineering and applied mathematics. Therefore, I would offer you an opportunity to address the comments, and probably submit it to engineering or applied/computational mathematics category after revision.

Journal Requirements:

2. Please upload a copy of Supporting Information File 1 which you refer to in your text on page 14.

Reviewers' comments:

Reviewer's Responses to Questions

**Comments to the Author**

1. Is the manuscript technically sound, and do the data support the conclusions?

Reviewer #1: Yes

Reviewer #2: Partly

2. Has the statistical analysis been performed appropriately and rigorously? 

Reviewer #1: Yes

Reviewer #2: N/A

3. Have the authors made all data underlying the findings in their manuscript fully available?

Reviewer #1: Yes

Reviewer #2: Yes

4. Is the manuscript presented in an intelligible fashion and written in standard English?

Reviewer #1: Yes

Reviewer #2: Yes

5. Review Comments to the Author

Reviewer #1: This manuscript presented a new formulation based on a consistent theoretical analysis of the NS equations. Numerical simulations are used to validate the proposed formula, showing that the new formula produces more accurate results than other existing models. This manuscript is well written and organized, I have some concerns as follows:

1. It seems that the manuscript is focusing on the development and validation of a new model, which is basically a formula, for the prediction of pressure drop. I feel it does not fit the current title "numerical investigation of ...". The numerical simulation is only a way to validate the proposed model.

2. The authors claimed that the proposed model can handle turbulence better with the new term, I would like to see more discussion in terms of this point.

3. The comparisons presented here are all assuming the presented numerical results as the accurate one, which is not convincing, I believe comparisons considering benchmark cases or data from others are necessary.

4. The Re is not defined in the manuscript.

5. The proposed model uses flow rate instead of velocity, why and what is the difference.

Reviewer #2: The work presents a new reduced-order model for pressure drop across the stenosis. This model technically sounds and is attractive.

Unfortunately, the design of this study is inappropriate. The authors apply their findings to the FFR calculations, i.e. the pressure drop evaluation in the stenosed coronary arteries. The authors compare their reduced order model with the 3D CFD flow in a single tube with rigid walls, sinus-like inflow velocity profile and zero gradient outflow boundary conditions. They demonstrate excellent agreement with this model. Unfortunately, the reference CFD model is entirely unrealistic. Essential features of the coronary circulation and FFR evaluation are the constant myocardium contractions, flexible walls, network structure of the vascular network, the change in the terminal resistance due to hyperemia due to the vasodilator administration. All these features are important but beyond the reference model. Thus it is not a surprise that the authors failed to reproduce the findings of the other works on virtual FFR evaluation.

I suggest the following todo list before the future submissions:

1) to study recent reviews on atherosclerosis modelling by a reduced-order approach;

2) to study recent works on virtual FFR evaluation by reduced order approaches, especially patient-specific modelling with (at least) acceptable statistical accuracy (A lot of contemporary works demonstrate suitable correlation with medical data.);

3) note that FFR is not the only parameter in decision making; several other coefficients have been introduced and studied recently;

4) check and compare the model's assumptions and realistic physiological conditions (It seems that near transitional turbulent regime hardly occurs in small coronary arteries with constantly collapsible terminals. Probably, the model would be valid for other locations of the stenosis (double-check the patient or general physiological data for that localisations before the next attempt) );

5) keep in mind that elastance of the vascular plays a significant role in haemodynamics.

6. PLOS authors have the option to publish the peer review history of their article (what does this mean?). If published, this will include your full peer review and any attached files.

Reviewer #1: No

Reviewer #2: No

---

## [Author Response · Author response to Decision Letter 0]

2 Jul 2021

We have addressed all the issues raised in the uploaded documents entitled: Response_to_Reviewers.docx and the Cover_letter.docx as instructed by the Editor. 

Our response included graphics such as figures and tables which are all included in the two documents. 

If any parts are not visible, please let us know at the email of the corresponding author in the manuscript.

---

## [Decision Letter · Decision Letter 1]

21 Jul 2021

PONE-D-21-04950R1

An improved reduced-order model for pressure drop across

arterial stenoses

PLOS ONE

Dear Dr. Lyras,

Thank you for submitting your manuscript to PLOS ONE. After careful consideration, we feel that it has merit but does not fully meet PLOS ONE’s publication criteria as it currently stands. Therefore, we invite you to submit a revised version of the manuscript that addresses the points raised during the review process.

We look forward to receiving your revised manuscript.

Kind regards,

Fang-Bao Tian

Academic Editor

PLOS ONE

Journal Requirements:

Additional Editor Comments (if provided):

Thank you for revising your work. As mentioned my one reviewer, the changes of the revised manuscript should reflect the discussions in the response letter. I would encourage the authors to do so, and then I am happy to recommend its publication in Plos One.

Reviewers' comments:

Reviewer's Responses to Questions

**Comments to the Author**

1. If the authors have adequately addressed your comments raised in a previous round of review and you feel that this manuscript is now acceptable for publication, you may indicate that here to bypass the “Comments to the Author” section, enter your conflict of interest statement in the “Confidential to Editor” section, and submit your "Accept" recommendation.

Reviewer #1: All comments have been addressed

Reviewer #2: All comments have been addressed

2. Is the manuscript technically sound, and do the data support the conclusions?

Reviewer #1: Yes

Reviewer #2: Yes

3. Has the statistical analysis been performed appropriately and rigorously? 

Reviewer #1: Yes

Reviewer #2: N/A

4. Have the authors made all data underlying the findings in their manuscript fully available?

Reviewer #1: Yes

Reviewer #2: Yes

5. Is the manuscript presented in an intelligible fashion and written in standard English?

Reviewer #1: Yes

Reviewer #2: Yes

6. Review Comments to the Author

Reviewer #1: The proposed questions are well addressed and I would like to recommend it to be published on Plos one.

Reviewer #2: Thank you for the detailed answer. The required reviews are not for me, but for the authors and for the readers. Unfortunately, a lot of the discussion in the reply was not revealed in the text. Yet, I suggest the changes are sufficient for accepting the paper for publication.

7. PLOS authors have the option to publish the peer review history of their article (what does this mean?). If published, this will include your full peer review and any attached files.

Reviewer #1: No

Reviewer #2: No

---

## [Author Response · Author response to Decision Letter 1]

6 Aug 2021

Thank you for the valuable comments. We have added a new section “Physiological and Clinical Relevance” towards the end of our Discussion in lines 561-586. In it, we bring together the issues explored in #3, #4 and #5 of our original response to reviewer 2 that were not explicitly included in the text before. As #1 and #2 were covered in our previous iteration, we believe that all relevant issues raised by the reviewers are now fully addressed in the text.

---

## [Editor Report · Decision Letter 2]

17 Sep 2021

An improved reduced-order model for pressure drop across

arterial stenoses

PONE-D-21-04950R2

Dear Dr. Lyras,

We’re pleased to inform you that your manuscript has been judged scientifically suitable for publication and will be formally accepted for publication once it meets all outstanding technical requirements.

Kind regards,

Fang-Bao Tian

Academic Editor

PLOS ONE
---

## [Editor Report · Acceptance letter]

24 Sep 2021

PONE-D-21-04950R2 

An improved reduced-order model for pressure drop across
arterial stenoses 

Dear Dr. Lyras:

I'm pleased to inform you that your manuscript has been deemed suitable for publication in PLOS ONE. Congratulations! Your manuscript is now with our production department. 

Kind regards, 

on behalf of

Dr. Fang-Bao Tian 

Academic Editor

PLOS ONE